

# The relationship of low-density lipoprotein cholesterol and all-cause or cardiovascular mortality in patients with type 2 diabetes: a retrospective study

Chin-Huan Chang[1], Shu-Tin Yeh[1], Seng-Wei Ooi[1], Chung-Yi Li[2,3,4] and Hua-Fen Chen[1,5]

[1] Department of Endocrinology, Far Eastern Memorial Hospital, New Taipei City, Taiwan
[2] Department of Public Health, College of Medicine, National Cheng Kung University, Tainan City, Taiwan
[3] Department of Public Health, College of Public Health, China Medical University, Taichung City, Taiwan
[4] Department of Healthcare Administration, College of Medical and Health Science, Asia University, Taichung City, Taiwan
[5] School of Medicine and Department of Public Health, College of Medicine, Fujen Catholic University, New Taipei City, Taiwan

Corresponding author
Hua-Fen Chen,
hfchen@mail.femh.org.tw

## ABSTRACT

**Background**. The optimal levels of low-density lipoprotein cholesterol (LDL-C) in patients with type 2 diabetes (T2D) are not currently clear. In this study, we determined the relationship between various mean LDL-C and all-cause or cardiovascular mortality risks in patients with T2D, stratifying by albumin level, age, sex, and antilipid medication use. We also evaluated the association of LDL-C standard deviation (LDL-C-SD) and all-cause and cardiovascular mortality by type of antilipid medication use.
**Methods**. A total of 46,675 T2D patients with a prescription for antidiabetic agents >6 months from outpatient visits (2003–2018) were linked to Taiwan's National Death Registry to identify all-cause and cardiovascular mortality. The Poisson assumption was used to estimate mortality rates, and the Cox proportional hazard regression model was used to assess the relative hazards of respective mortality in relation to mean LDL-C in patient cohorts by albumin level, age, sex, and antilipid use adjusting for medications, comorbidities, and laboratory results. We also determined the overall, and anti-lipid-specific mortality rates and relative hazards of all-cause and cardiovascular mortality associated with LDL-C-SD using the Poisson assumption and Cox proportional hazard regression model, respectively.
**Results**. All-cause and cardiovascular mortality rates were the lowest in T2D patients with a mean LDL-C > 90-103.59 mg/dL in the normal albumin group ($\geq$ 3.5 g/dL). Compared to T2D patients with a mean LDL-C > 90–103.59 mg/dL, those with a mean LDL-C $\leq$ 77 mg/dL had an elevated risk of all-cause mortality in both the normal and lower albumin groups. T2D patients with a mean LDL-C $\leq$ 90 and > 103.59–119 mg/dL had relatively higher risk of cardiovascular mortality in the normal albumin group, but in the lower albumin group (<3.5 g/dL), any level of mean LDL-C $\leq$ 119 mg/dL was not significantly associated with cardiovascular mortality. Increased risks of all-cause and cardiovascular mortality were observed in patients with a mean LDL-C $\leq$ 77 mg/dL in both sexes and in all age groups except in those aged <50 years, a lower mean LDL-C was not associated with cardiovascular mortality. Similarly, patients with an LDL-C-SD <10th and > 90th percentiles were associated with significant risks of all-cause and

cardiovascular mortality. In statin users, but not fibrate users, lower and higher levels of mean LDL-C and LDL-C-SD were both associated with elevated risks of all-cause and cardiovascular mortality.

**Conclusions**. The optimal level of LDL-C was found to be >90–103.59 mg/dL in T2D patients. Lower and higher levels of mean LDL-C and LDL-C-SD were associated with all-cause and cardiovascular mortality, revealing U-shaped associations. Further studies are necessary to validate the relationship between optimal LDL-C levels and all-cause and cardiovascular mortality in patients with diabetes.

## INTRODUCTION

Lowering low-density lipoprotein cholesterol (LDL-C) levels has been shown to reduce both all-cause and cardiovascular mortality in some (*Pedersen et al., 2004*; *Tonkin et al., 1998*; *Yusuf, 2002*; *Athyros et al., 2002*) but not all trials (*Trial, 2002*; *Amarenco et al., 2006*; *Armitage et al., 2010*; *Pedersen et al., 2005*). Although diabetes is regarded as a cardiovascular disease risk equivalent (*Kuusisto & Laakso, 2013*), some clinical trials of patients with diabetes on lipid-lowering medications showed no difference in mortality (*Colhoun et al., 2004*; *Knopp et al., 2006*). In a meta-analysis and meta-regressions, more intensive LDL-C lowering was associated with a greater reduction in both all-cause and cardiovascular mortality risk only when baseline LDL-C was more than 100 mg/dL (*Navarese et al., 2018*). In one general population-based study (*Johannesen et al., 2020*) and a separate systemic review (*Ravnskov et al., 2016*), low levels of LDL-C were associated with increased risk of mortality, and the lowest risk of all-cause mortality was found at an LDL-C concentration of 140 mg/dL (*Johannesen et al., 2020*). However, the general population-based study (*Johannesen et al., 2020*) obtained only a single measurement of baseline LDL-C in their analysis rather than multiple LDL-C measurements with a long-term longitudinal follow-up in real word practice.

To the best of our knowledge, the optimal levels of LDL-C for all-cause or cardiovascular mortality in patients with diabetes stratified by albumin level, age, and sex have not been extensively studied. Some previous studies have also shown that visit-to-visit variability (VVV) of lipid measurements may affect all-cause and cardiovascular mortality (*Liu et al., 2020*; *Wan et al., 2020*), though one other study did not find these associations (*Lee et al., 2021*). High- to moderate- intensity statin therapy is recommended for diabetes patients with and without atherosclerotic cardiovascular disease (*Committee, 2021*), but the long-term clinical association between various antilipid medications and mortality in different percentiles of mean and VVV of LDL-C are currently unknown. Several health-related conditions such as malnutrition, renal disease, and other chronic illnesses, including subclinical diseases, might be related to low cholesterol level, and *Corti et al. (1997)* demonstrated that the inverse association between mortality and total cholesterol (TC) disappeared after adjusting for markers of poor health including albumin level. In another

analysis of patients with coronary artery disease, lower baseline serum albumin and TC indicated a higher risk of major adverse cardiac events (*Yao et al., 2022*). Albumin is closely related to the metabolism, biosynthesis, and transport of cholesterol (*Sankaranarayanan et al., 2013*; *Kumar et al., 2018*), and serum albumin levels may affect the prognosis of different levels of LDL-C in patients with diabetes.

The main goal of disease prevention is to prolong life, and all-cause and cardiovascular mortality are easily defined outcomes, and the least subject to bias (*Ravnskov et al., 2016*). The purpose of this study was to determine the relationship between mean LDL-C levels and all-cause and cardiovascular mortality in patients with type 2 diabetes (T2D) treated at Far Eastern Memorial Hospital (FEMH) from Jan 1, 2003 to Dec 31, 2018. We performed the statistical analyses stratifying by albumin level, age, and sex to see whether albumin levels and demographic factors would affect the rates and risks of all-cause and cardiovascular mortality. Another aim of our study was to estimate the risks of all-cause and cardiovascular mortality in association with different percentiles of VVV of LDL-C in our patients. We chose standard deviation (SD) to represent the VVV of LDL-C (LDL-C-SD) metrics in our study because it captures the data variability of both short or long-time intervals and is less likely to be influenced by extreme observations (*Hsu et al., 2021*). We also evaluated the influence of various types of lipid-lowering medications and different levels of mean LDL-C and LDL-C-SD on all-cause and cardiovascular mortality.

## MATERIALS & METHODS

### Study design and subjects

This study used a cohort design to assess the all-cause and cardiovascular mortality of T2D patients in relation to various percentiles of mean LDL-C and LDL-C-SD at FEMH, a tertiary medical center in Northern Taiwan. Twelve endocrinologists, together with cardiologists, neurologists, nephrologists, and family medicine physicians take care of 96% of outpatient patients with diabetes. The total number of patients with diabetes in FEMH was the fourth largest among all medical centers in Taiwan (*The National Health Insurance Administration & the Ministry of Health and Welfare Website, 2021*).

Since Jan 1, 2001, FEMH has used an electronic medical database of outpatient visits that includes information about patient age, sex, hospital chart number, personal identification number (PIN), dates of admission and discharge, length of hospital stay, up to six ICD diagnosis codes, prescribed medications, and laboratory reports, including point-of-care (POC) glucometer data. This study was approved by the Research Ethics Review Committee of FEMH (110276-F) with exemption from informed consent.

We identified 74,888 patients with ICD-9-CM: 250.xx or ICD-10-CM: E10 or E11 in out-patient visits between Jan. 1, 2003 to Dec. 31, 2018. We excluded 4,213 patients with no prescription of oral or parenteral antidiabetic agents, 23,530 patients with a total outpatient visit duration at FEMH less than 6 months, 470 patients with type 1 diabetes; the final study cohort consisted of 46,675 T2D patients. The index date of each patient was the date of first oral or parenteral antidiabetic agent use at FEMH during the study period.

## Follow-up, covariates, and study endpoints

If the patients did not encounter mortality during the study period, and their outpatient visits were detected beyond Dec. 31, 2018, they were censored at the end of the study period (*i.e.,* Dec 31, 2018). For other patients, the dates of last outpatient visit in FEMH were set to be their censored dates.

The age of each study subject was determined using the difference between the index date and the date of birth. Similar to the study design of our previous reports (*Ooi et al., 2022*; *Yeh et al., 2022*), we collected all information on oral antidiabetic agents, insulin, antihypertensive, and antilipid medications (statins and fibrates) between the index date and the end of follow-up, which is either death or censoring. Various cardiovascular risk factors (coronary artery disease (ICD-9-CM: 410, 411, 413, 414 or ICD-10-CM: I20-I25), heart failure (ICD-9-CM: 428 or ICD-10-CM: I50), and cerebrovascular disease (ICD-9-CM: 433-436, 437.0, 437.1 or ICD-10-CM: I63-I66, I67.2, I67.3, I67.6, I67.81, I67.82, I67.9)) retrieved from the outpatient medical records between the index date and the end of follow-up were considered as potential confounders (*Glasheen, Renda & Dong, 2017*).

We also collected each patient's glycated hemoglobin (HbA1c), fasting plasma glucose (FPG), TC, triglycerides (TG), high-density lipoprotein cholesterol (HDL-C), LDL-C, creatinine (Cr), albumin, and hemoglobin (Hb) levels during the study period. The eGFR was assessed using the Modification of Diet in Renal Disease (MDRD) Study equation and the Cr standardized to reference methods (*Levey et al., 2007*). We selected the highest HbA1c, FPG, eGFR, TC, TG, HDL-C, and LDL-C of each quarter for each patient's representative HbA1c, FPG, eGFR, TC, TG, HDL-C, LDL-C, and then averaged the annual value of that specific year. Mean of the annual mean during the whole study period was computed thereafter. Hyperglycemia was evaluated using the mean of the annual mean HbA1c from the index date to the last year of follow-up. The measurement of HbA1c assay in FEMH is certified by the National Glycohemoglobin Standardization Program (NGSP). Each year's lowest albumin and Hb were collected to calculate the mean of the annual mean of each to prevent overestimation of those values after replacement. Mean albumin levels were grouped into ≥3.5, 3.0−3.4, 2.5−2.9, and <2.5 g/dL (*Liu et al., 2021a*; *Liu et al., 2021b*). Hypoglycemia was identified in patients with fasting or postprandial plasma or POC glucose values <70 mg/dL on any occasion during the study period (*Draznin et al., 2022*).

Hyperlipidemia was evaluated using the mean of the annual mean LDL-C and intra-individual LDL-C-SD from the index date to the last year of follow-up. These mean LDL-C values were then categorized into ≤10th (≤77 mg/ dL), >10th–≤25th (>77–90 mg/dL), >25th–≤50th (>90–103.59 mg/dL), >50th–≤75th (>103.59–119 mg/dL), >75th–≤90th (>119–135.59 mg/dL), and >90th (>135.59 mg/dL) percentiles. We computed each patient's annual LDL-C-SD separately, and the overall mean LDL-C-SD during the whole study period was calculated from the average of each of the annual values. Patients with T2D were then stratified into ≤ 10th, >10th–≤25th, >25th–≤50th, >50th–≤75th, >75th–≤90th, and >90th percentiles of LDL-C-SD measurements. We linked this electronic database to Taiwan's National Death Registry, which contains information on age, sex, dates, and

causes of death, with the unique PIN. There is a mandatory registration of all deaths in Taiwan, and all death certificates must be completed by physicians, so the mortality registry is generally accurate and complete (*Ho et al., 2010*; *Lu, Lee & Chou, 2000*).

The study endpoints were all-cause mortality and cardiovascular mortality. Cardiovascular mortality was identified if the patient's cause of death was coded with ICD-9-CM: 390-392, 393-398, 410-414, 420-429, 430-438 or ICD-10-CM: I01-I02.0, I05-I09, I20-25, I27, I30-I52, I60-I69.

## Statistical analysis

All-cause and cardiovascular mortality rates of mean LDL-C and LDL-C-SD were evaluated with person-years as the denominator using the Poisson assumption. We determined the independent association of mean LDL-C percentile with the relative hazards of all-cause or cardiovascular mortality using a Cox proportional hazard regression model. We also assessed the mortality rates and relative hazards of each mean LDL-C percentile in relation to all-cause or cardiovascular mortality according to: level of mean albumin ($\geq 3.5$, $3.0-3.4$, $2.5-2.9$, and $<2.5$ g/dL), sex (men *vs.* women), age group ($<50$, 50-69, $>69$ years), and type of antilipid use (statins *vs.* fibrates). Patients with a mean LDL-C $>90$–103.59 mg/dL ($>$25th–$\leq$50th percentile) during the follow-up period were used as the reference group in each stratification. The independent association of each stratification of LDL-C-SD with the rates and relative hazards of all-cause and cardiovascular mortality were also determined using a Cox proportional hazard regression model with the $>$25th–$\leq$50th percentile of LDL-C-SD used as the reference range. We also explored the rates and relative hazards of all-cause or cardiovascular mortality of different percentiles of LDL-C-SD between statin and fibrates use.

In model 1, the regression models were adjusted for, age, and sex. In model 2, we adjusted for prescribed antidiabetic, antihypertensive, and antilipid medications in addition to the variables in model 1. We then added selected cardiovascular risk factors (coronary artery disease, heart failure, and cerebrovascular disease) together with laboratory results such as mean HbA1c, FPG, TC, TG, HDL-C, eGFR, albumin, and Hb levels to the regression models in model 3. All statistical analyses were performed with SAS (version 9.4; SAS Institute, Cary, NC). A *p*-value $<0.05$ was considered statistically significant. Survival plots of all-cause and cardiovascular mortality related to mean LDL-C were depicted using the Kaplan–Meier method.

## RESULTS

### Baseline patient characteristics

Table 1 presents the baseline characteristics of the study subjects by mean LDL-C percentile. Patients aged $<69$ years were more likely to have a mean LDL-C $>77$ mg/dL than those aged $>69$ years. More men had their mean LDL-C kept $\leq 77$ mg/dL, while more women had higher mean LDL-C $>119$ mg/dL.

There were no considerable differences in antidiabetic medication, antihypertensive medication, or fibrates use between the different percentiles of mean LDL-C. However, statin prescriptions increased with mean LDL-C percentile. Coronary artery disease, heart
**Table 1 Characteristics of participating patients with type 2 diabetes by mean low-density lipoprotein cholesterol percentile.**

| Mean LDL-C (mg/dL) in percentile[a,b] | Mean LDL (%) | | | | | |
|---|---|---|---|---|---|---|
| | <10th | >10th–≤25th | >25th–≤50th | >50th–≤75th | >75th–≤90th | >90th |
| | ≤77 | >77–90 | >90–103.59 | >103.59–119 | >119–135.59 | >135.59 |
| Total | 4,317 | 6,451 | 10,710 | 10,842 | 6,336 | 4,295 |
| **General characteristics** | | | | | | |
| Mean age (±SD) | 61.52 ± 13.80 | 59.82 ± 12.99 | 58.88 ± 12.33 | 57.67 ± 12.59 | 56.83 ± 12.95 | 56.02 ± 12.95 |
| <50 years | 21.00% | 21.71% | 23.21% | 26.06% | 28.48% | 30.15% |
| 50–69 years | 49.04% | 55.45% | 57.26% | 57.01% | 55.28% | 55.35% |
| >69 years | 29.96% | 22.84% | 19.53% | 16.94% | 16.25% | 14.50% |
| Sex | | | | | | |
| Women | 40.47% | 44.14% | 46.42% | 47.32% | 48.58% | 49.41% |
| Men | 59.53% | 55.86% | 53.58% | 52.68% | 51.42% | 50.59% |
| **Antidiabetic medications** | | | | | | |
| Oral hypoglycemic agents | 94.86% | 96.98% | 97.25% | 97.86% | 97.51% | 96.39% |
| Insulin | 36.44% | 33.81% | 32.76% | 29.63% | 28.96% | 33.99% |
| **Antihypertensives** | 87.61% | 86.92% | 85.14% | 82.96% | 80.76% | 77.63% |
| **Antilipids** | | | | | | |
| Statin | 51.15% | 66.19% | 72.71% | 74.49% | 75.54% | 81.37% |
| Fibrates | 22.19% | 23.25% | 22.89% | 22.75% | 22.46% | 20.21% |
| **Comorbidities and Complications** | | | | | | |
| Coronary artery disease | 39.82% | 36.77% | 33.10% | 30.13% | 28.19% | 27.54% |
| Heart failure | 18.21% | 13.58% | 12.11% | 11.03% | 10.76% | 11.92% |
| Cerebrovascular Disease | 15.71% | 16.46% | 16.92% | 15.94% | 13.70% | 12.99% |
| Hypoglycemia | 19.94% | 23.72% | 22.25% | 19.45% | 15.61% | 13.36% |
| **Laboratory results** | | | | | | |
| **Mean (SD)** | | | | | | |
| HbA1c (%) | 7.67 ± 1.42 | 7.59 ± 1.20 | 7.65 ± 1.19 | 7.77 ± 1.26 | 7.99 ± 1.43 | 8.44 ± 1.72 |
| FPG (mg/dL) | 149.44 ± 45.82 | 145.79 ± 38.25 | 145.82 ± 36.02 | 147.60 ± 37.11 | 152.19 ± 41.84 | 164.26 ± 50.32 |
| TC (mg/dL) | 145.74 ± 35.25 | 158.83 ± 21.57 | 169.32 ± 18.42 | 183.23 ± 19.02 | 198.40 ± 20.16 | 226.13 ± 33.68 |
| TG (mg/dL) | 194.76 ± 258.54 | 166.21 ± 122.64 | 161.18 ± 100.46 | 168.17 ± 96.97 | 180.29 ± 101.43 | 201.94 ± 139.37 |
| HDL-C (mg/dL) | 45.26 ± 15.34 | 46.60 ± 13.49 | 46.71 ± 12.47 | 46.52 ± 11.86 | 46.37 ± 11.23 | 47.66 ± 12.00 |
| LDL-C (mg/dL) | 65.39 ± 10.81 | 84.41 ± 3.66 | 96.99 ± 3.85 | 110.82 ± 4.47 | 126.22 ± 4.67 | 155.78 ± 23.19 |
| eGFR (ml/min/m$^2$) | 53.43 ± 12.65 | 54.80 ± 10.82 | 55.55 ± 10.01 | 55.79 ± 9.82 | 55.43 ± 10.43 | 53.63 ± 12.73 |
| Hb (g/dL) | 12.28 ± 2.28 | 12.80 ± 2.11 | 13.00 ± 2.02 | 13.09 ± 2.05 | 13.09 ± 2.11 | 12.84 ± 2.33 |
| ≥11 | 70.09% | 79.40% | 83.64% | 84.23% | 83.29% | 76.39% |
| <11 | 29.91% | 20.60% | 16.36% | 15.77% | 16.71% | 23.61% |
| Albumin (g/dL) | 3.70 ± 0.67 | 3.90 ± 0.63 | 3.99 ± 0.62 | 4.02 ± 0.64 | 3.97 ± 0.66 | 3.80 ± 0.72 |
| ≥3.5 | 64.43% | 76.20% | 80.87% | 81.61% | 78.83% | 68.50% |
| 3.0–3.4 | 20.16% | 14.92% | 11.73% | 11.06% | 12.70% | 17.60% |
| 2.5–2.9 | 11.02% | 6.20% | 5.22% | 5.14% | 5.57% | 8.82% |
| <2.5 | 4.39% | 2.68% | 2.18% | 2.19% | 2.90% | 5.08% |

**Table 1** (*continued*)

| Outcome | | | | | | |
|---|---|---|---|---|---|---|
| All-cause mortality | 1,630 (37.76%) | 1,596 (24.74%) | 2,237 (20.89%) | 2,364 (21.80%) | 1,492 (23.55%) | 1,322 (30.78%) |
| Cardiovascular mortality | 296 (6.86%) | 296 (4.59%) | 444 (4.15%) | 503 (4.64%) | 306 (4.83%) | 278 (6.41%) |
| Mean follow-up period (year) | 6.17 ± 4.27 | 7.27 ± 4.55 | 7.58 ± 4.60 | 7.18 ± 4.55 | 6.41 ± 4.44 | 5.51 ± 4.17 |

**Notes.**
[a]Data are % or mean (±SD).
[b]eGFR, estimated glomerular filtration rate; FPG, fasting plasma glucose; Hb, hemoglobin; HbA1c, glycated hemoglobin; HDL-C, high-density lipoprotein cholesterol; LDL-C, low-density lipoprotein cholesterol; SD, standard deviation; TC, total cholesterol; TG, triglycerides.

failure, cerebrovascular disease, and hypoglycemia were more prevalent in patients with lower percentiles of mean LDL-C.

Patients with higher percentiles of mean LDL-C had higher mean HbA1c, FPG, TC, TG, and HDL-C. There was no difference in mean eGFR, albumin, or Hb across each percentile of mean LDL-C, but there was a higher prevalence of low albumin (<2.5 g/dL) and anemia (Hb <11 g/dL) in the lowest mean LDL-C percentile.

During follow-up (a mean of around seven years), the highest percentage of all-cause and cardiovascular mortality was found in those with a mean LDL-C <10th percentile, ≤77 mg/dL (37.76% and 6.86%, respectively), while those with a mean LDL-C between >90–103.59 mg/dL (>25th–≤50th percentile) had the lowest proportion of all-cause and cardiovascular mortality (20.89% and 4.15%, respectively).

## Overall and mean albumin level-specific rates and relative hazards of all-cause mortality by mean LDL-C percentile

The overall all-cause mortality was lowest in T2D patients with a mean LDL-C between >90–103.59 mg/dL (25th–50th percentile; 27.58/1,000 patient-year (PY); 95% confidence interval (CI) [26.44–28.72]) while the highest mortality was observed in those with a mean LDL-C ≤77 mg/dL (<10th percentile; 61.15 PY; 95% CI [58.18–64.12]; Table S1). The survival plot of all-cause mortality by mean LDL-C percentile is presented in Fig. 1. Compared to those in the 25th–50th percentile of mean LDL-C, those with higher and lower values of mean LDL-C had a higher hazard ratio (HR) of all-cause mortality (Fig. 2). Adjusting for medications, comorbidities, and laboratory results attenuated the HRs but they were still persistently and statistically significant in all other percentiles of LDL-C. The highest and second highest HRs of all-cause mortality were observed in patients with a mean LDL-C >135.59 mg/dL (>90th percentile; HR: 1.52; 95% CI [1.38–1.68]) and ≤77 mg/dL (<10th percentile; HR: 1.47; 95% CI [1.35–1.59]) in model 3 (Table S1). Because there was a significant interaction between mean LDL-C and mean albumin ($P < 0.0001$), we performed a stratified analysis to evaluate the albumin-specific rates and relative hazards of all-cause mortality by mean LDL-C percentile (Table S1). In all albumin groups, the lowest all-cause mortality rate was observed in those in the 25th–50th percentile of mean LDL-C (>90–103.59 mg/dL), but mortality rates in the lower albumin group (<3.5 g/dL) were 3–4 times higher than those in normal albumin group (≥3.5 g/dL). The highest mortality was noted in patients with mean LDL-C levels >90th percentile (>135.59 mg/dL) in the normal albumin group (≥3.5 g/dL; 47.38/1,000 PY) while in the

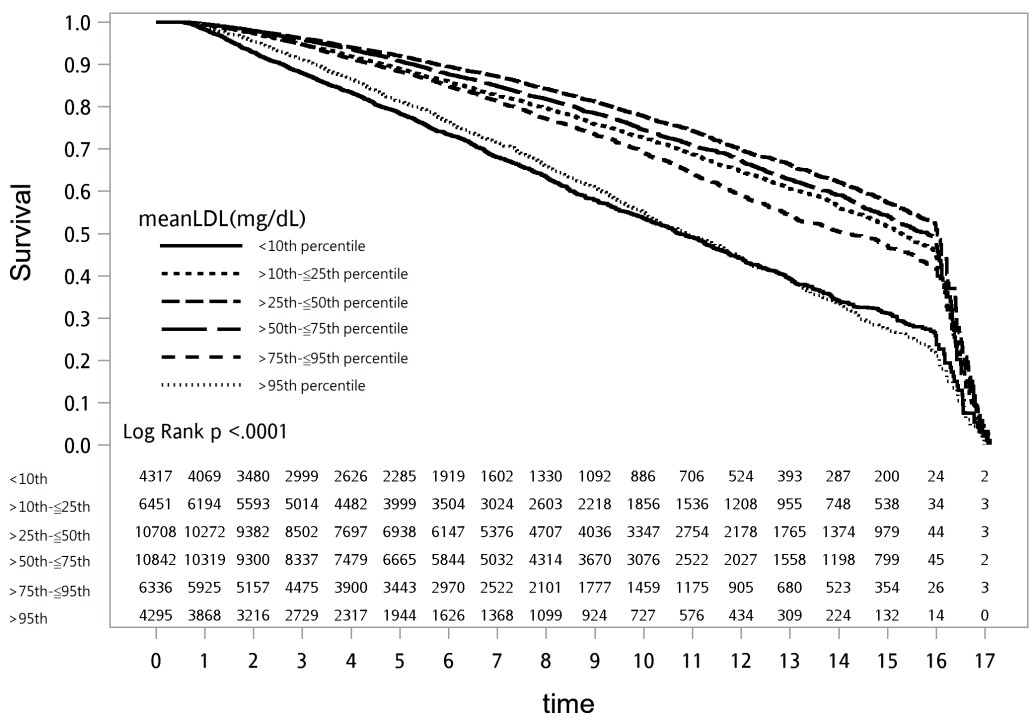

**Figure 1** Survival plot of mean low-density lipoprotein cholesterol (LDL-C)and all-cause mortality.

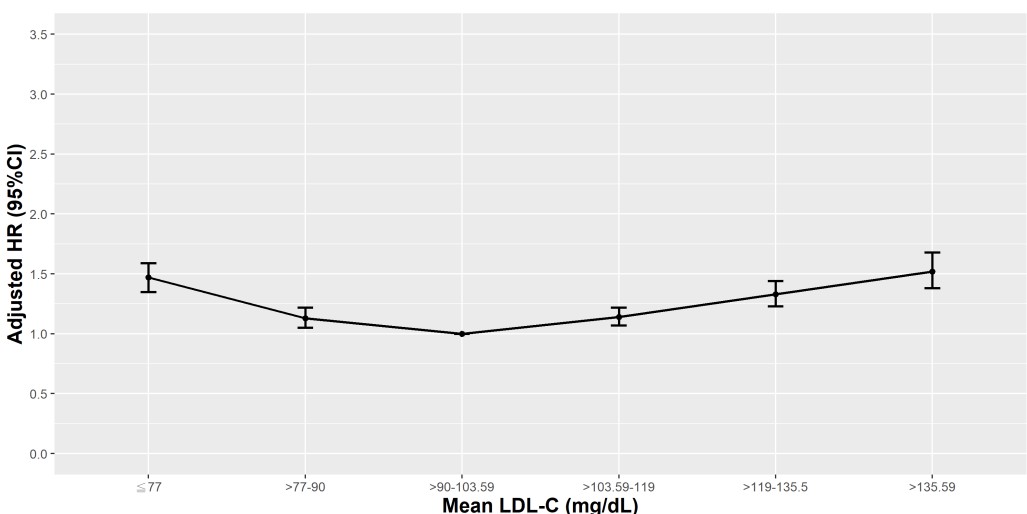

**Figure 2** Hazard plot of mean low-density lipoprotein cholesterol (LDL-C) and all-cause mortality.

lower albumin group (<3.5 g/dL), the highest mortality rates were consistently found in those with a mean LDL-C <10th percentile (≤77 mg/dL).

Compared to those in the 25th–50th percentile of mean LDL-C, those in the <25th (<90 mg/dL) and >90th percentile of mean LDL-C (>135.59 mg/dL) had higher HRs of

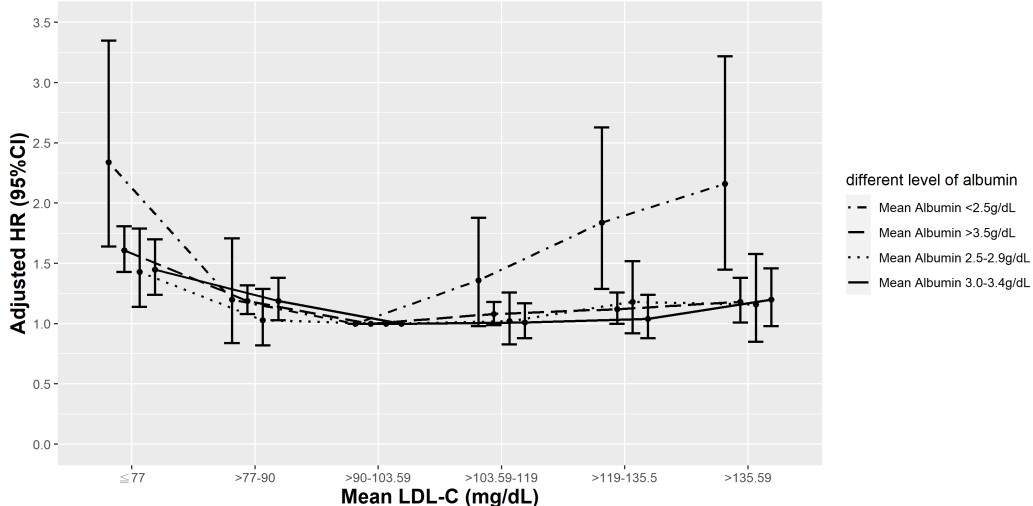

**Figure 3** Hazard plot of mean low-density lipoprotein cholesterol (LDL-C) and all-cause mortality with different albumin levels.

all-cause mortality, even after adjusting for comorbidities and laboratory results in model 3 in the normal albumin group ($\geq$3.5 g/dL). However, in T2D patients with a mean albumin 3.0−3.4 g/dL or 2.5−2.9 g/dL, elevated risks of all-cause mortality were observed in those with a mean LDL-C <25th percentile (<90 mg/dL) and <10th percentile ($\leq$77 mg/dL), respectively. Increased HRs observed at higher mean LDL-C levels became inconsequential after risk factor adjustments. In patients with the lowest mean albumin (<2.5 g/dL), a mean LDL-C in the <10th or >75th percentile ($\leq$77 and >119 mg/dL) significantly increased the risk of all-cause mortality, even after adjusting for all other covariates in model 3 (Fig. 3).

## Overall and mean albumin level-specific rates and relative hazards of cardiovascular mortality by mean LDL-C percentile

The lowest cardiovascular mortality was found in patients with a mean LDL-C >90–103.59 mg/dL (5.47/1,000 PY; 95% CI [4.96–5.98]). However, the most elevated cardiovascular mortality was seen in those with an LDL-C >90th percentile (>135.59 mg/dL; 11.74/1,000 PY) followed by those with a mean LDL-C $\leq$77 mg/dL (11.10/1,000 PY). The survival plot of all-cause mortality by mean LDL-C percentile is presented in Fig. 4. Compared to those in the 25th–50th percentile of mean LDL-C, those with a mean LDL-C $\leq$77 mg/dL (<10th percentile) or >103.59 mg/dL (>50th percentile) had an increased risk of cardiovascular mortality, even after adjusting for medications, comorbidities, and laboratory results (Fig. 5 and Table S2).

There was a significant interaction between mean LDL-C and mean albumin ($P$ < 0.0121), so we performed a stratified analysis to evaluate the albumin-specific rates and relative hazards of cardiovascular mortality by mean LDL-C percentile (Table S2). The lowest rate of cardiovascular mortality was seen in patients in the 25th–50th percentile of mean LDL-C and with a mean albumin $\geq$3.5 g/dL (4.71/1,000 PY), and the

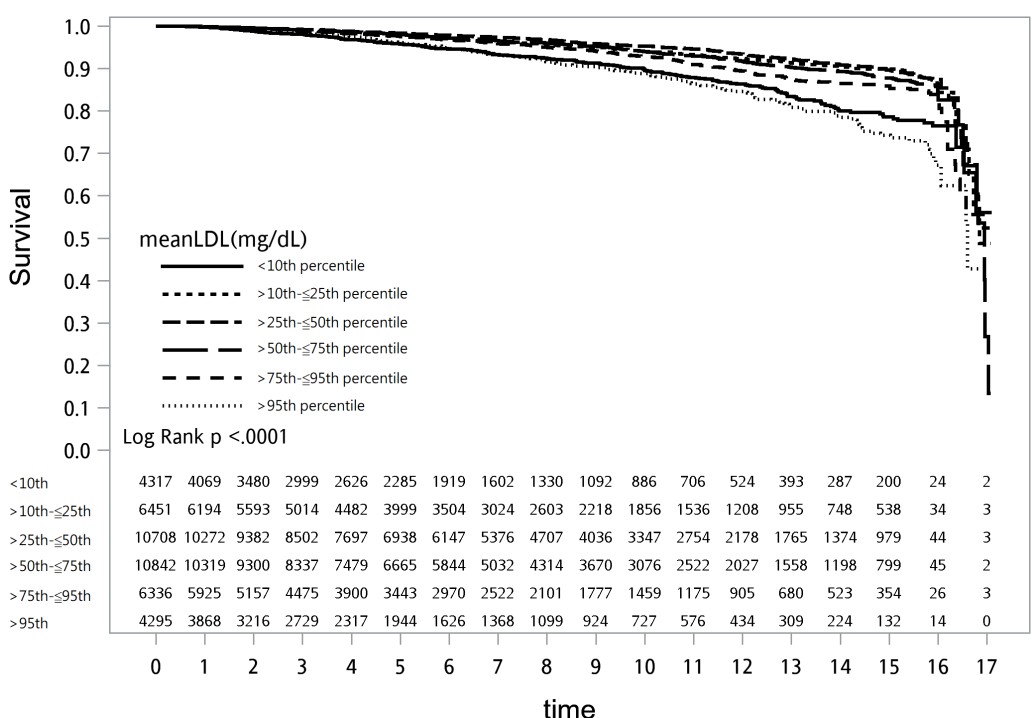

**Figure 4  Survival plot of mean low-density lipoprotein cholesterol (LDL-C) and cardiovascular mortality.**

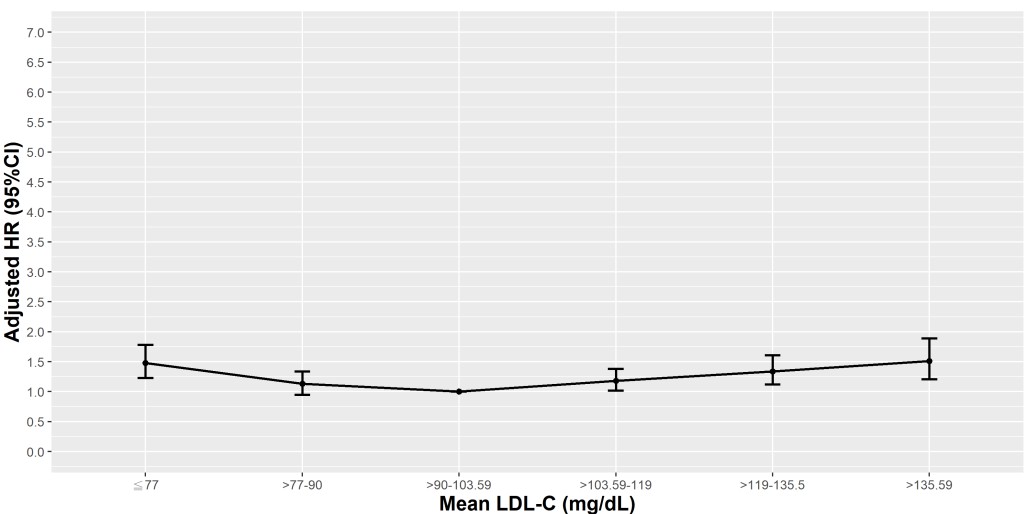

**Figure 5  Hazard plot of mean low-density lipoprotein cholesterol (LDL-C) and cardiovascular mortality.**

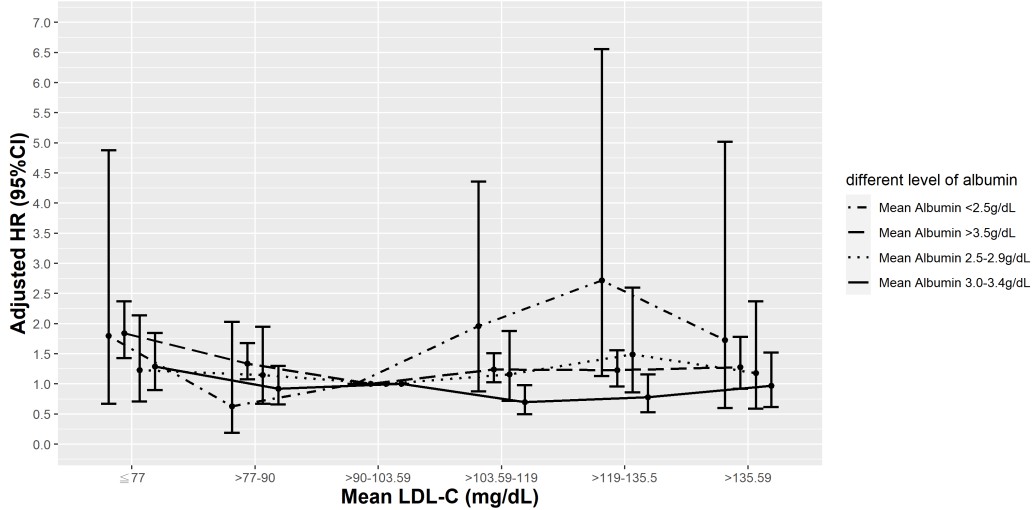

**Figure 6** **Hazard plot of mean low-density lipoprotein cholesterol (LDL-C) and cardiovascular mortality with different albumin levels.**

cardiovascular mortality rates were higher in those with lower albumin group (<3.5 g/dL). In the normal albumin group (≥3.5 g/dL), compared to patients with a mean LDL-C in the 25th–50th percentile (>90–103.59 mg/dL), patients with a mean LDL-C <25th percentile (≤90 mg/dL) or in the 50th–75th percentile (>103.59–119 mg/dL) had higher risks of cardiovascular mortality after adjusting for coronary risk factors and laboratory results. In the lower albumin group (<3.4 g/dL), mean LDL-C did not significantly change cardiovascular mortality risk after adjusting for risk factors, except in patients with a mean albumin <2.5 g/dL, and a mean LDL-C in the 75th–90th percentile (>119–135.59 mg/dL) who had higher HRs of cardiovascular mortality (Fig. 6).

## Sex-specific rates and relative hazards of all cause and cardiovascular mortality by mean LDL-C percentile

We further analyzed the sex-specific rates and relative hazards of all-cause and cardiovascular mortality by mean LDL-C percentile (Table S3). The *P* values for the interaction of mean LDL-C with sex for all-cause and cardiovascular mortality were <0.0001 and 0.0003, respectively. Irrespective of sex, the lowest all-cause and cardiovascular mortality rates were observed in patients with a mean LDL-C in the 25th–50th percentile (>90–103.59 mg/dL). The highest all-cause and cardiovascular mortality rates were noted in those with a mean LDL-C <10th percentile (≤77 mg/dL), except in men, the highest cardiovascular mortality rates were detected in male patients with a mean LDL-C >90th percentile (>135.5 mg/dL).

Compared to female patients with a mean LDL-C in the 25th–50th percentile (>90–103.59 mg/dL), female patients with a mean LDL-C <25th percentile or >50th percentile (<90 - >103.59 mg/dL) had higher HRs of all-cause mortality and female patients with a mean LDL-C <10th percentile or >75th percentile (<77 and >119 mg/dL) had higher HRs
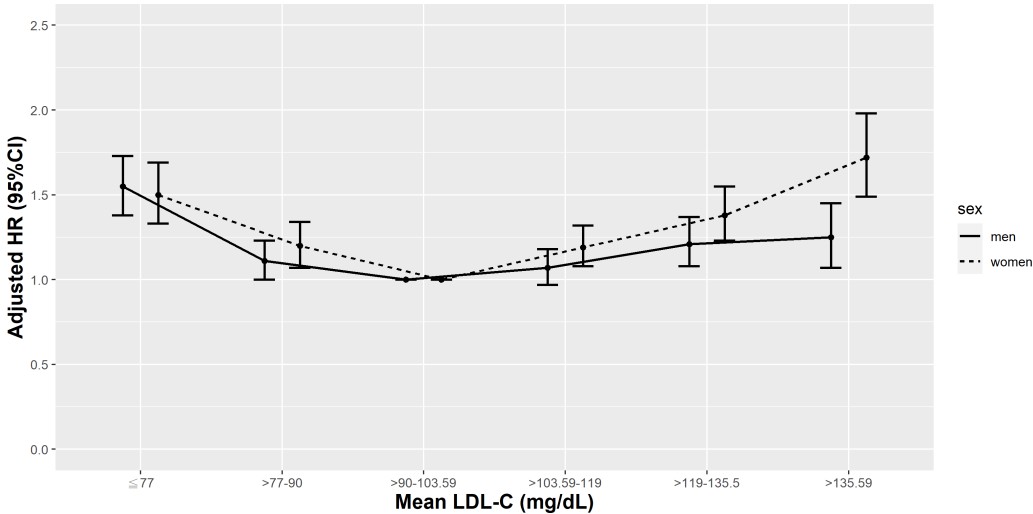

**Figure 7  Hazard plot of mean low-density lipoprotein cholesterol (LDL-C) and all-cause mortality with different sex.**

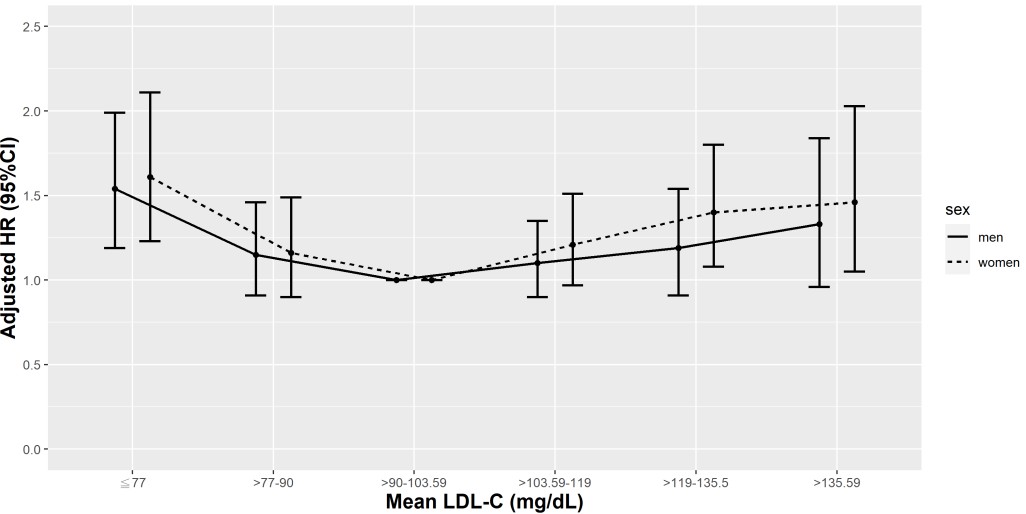

**Figure 8  Hazard plot of mean low-density lipoprotein cholesterol (LDL-C) and cardiovascular mortality with different sex.**

of cardiovascular mortality. However, in male T2D patients, only those with a mean LDL-C <10th or >75th percentile (<77 or >119 mg/dL) had higher risks of all-cause mortality and only men with a mean LDL-C <10th percentile (<77 mg/dL) had higher risks of cardiovascular mortality (Figs. 7 & 8).

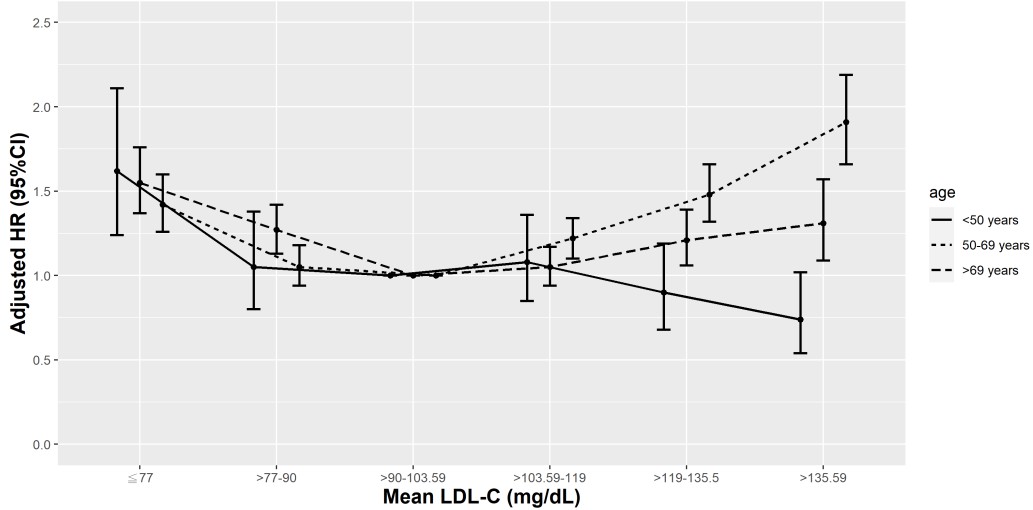

**Figure 9** Hazard plot of mean low-density lipoprotein cholesterol (LDL-C) and all-cause mortality with different age.

## Age-specific rates and relative hazards of all-cause and cardiovascular mortality by mean LDL-C percentile

Generally, lower all-cause and cardiovascular mortality rates were detected in those with a mean LDL-C in the 10th–75th percentile (>77–<135.5 mg/dL) and the highest all-cause and cardiovascular mortality rates were observed in patients with a mean LDL-C either <10th or >90th percentile (<77 or >135.5 mg/dL) in all age groups. (*P* values for the interaction of mean LDL-C with age for all-cause and cardiovascular mortality were 0.0367 and 0.8022, respectively; Table S4).

In T2D patients aged <50 years, only those with a mean LDL-C <10th percentile (≤77 mg/dL) had more elevated risks of all-cause mortality. Conversely, there were no increased risks of cardiovascular mortality observed in this age group at any level of mean LDL-C, after adjusting for medications, comorbidities, and laboratory results (Figs. 9 & 10). In T2D patients aged 50-69 years, those with a mean LDL-C <10th or >50th percentile (≤77 and >103.59 mg/dL) had higher risks of all-cause and cardiovascular mortality compared to those with a mean LDL-C in the 25th–50th percentile (>90–103.59 mg/dL), after adjusting for risk factors in model 3. However, in elderly patients aged >69 years, those with a mean LDL-C <25th or >75th percentile (≤90–>119 mg/dL) had increased risks of all-cause mortality and those with a mean LDL-C <10th percentile (≤ 77 mg/dL) had increased risks of cardiovascular mortality, after adjusting for various covariates.

## Anti-lipid-specific rates and relative hazards of all-cause and cardiovascular mortality by mean LDL-C percentile

Among statin and fibrates users, the lowest all-cause and cardiovascular mortality rates were noted in those with a mean LDL-C >90–103.59 mg/dL (25th–50th percentiles) and >77–90 mg/dL (10th–25th percentile), respectively. The most elevated rates of all-cause

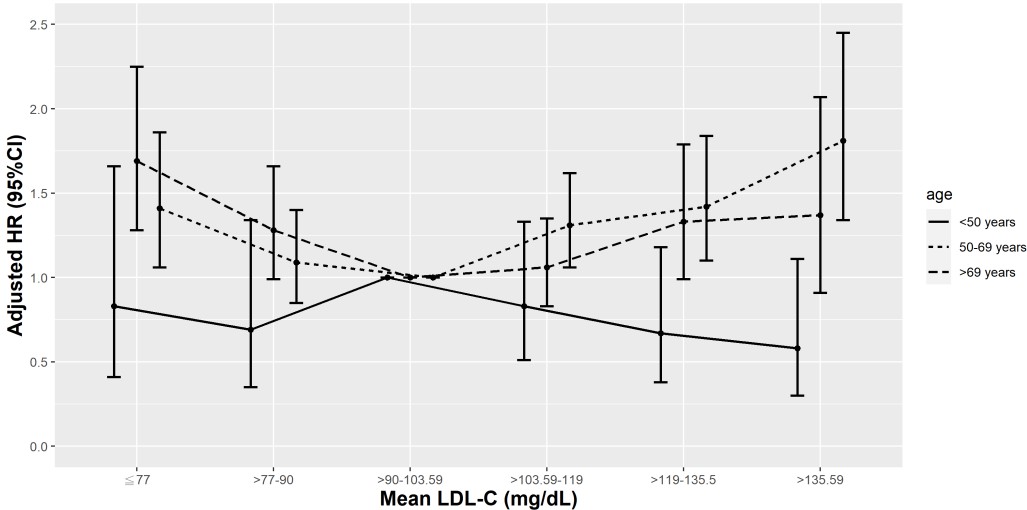

**Figure 10** Hazard plot of mean low-density lipoprotein cholesterol (LDL-C) and cardiovascular mortality with different age.

and cardiovascular mortality were consistently observed in those with the highest mean LDL-C (>135.59 mg/dL) (Table S5).

Compared to patients with a mean LDL-C in the 25th–50th percentile (90–103.59 mg/dL), statin users with a lower or higher mean LDL-C had significant risks of all-cause and cardiovascular mortality. In fibrates users, a lower or higher mean LDL-C (≤77 and >119 mg/dL was also associated with elevated all-cause mortality risk, but only fibrates users with a mean LDL-C >119 mg/dL had significant risks of cardiovascular mortality.

## Overall and antilipid-specific rates and relative hazards of all-cause and cardiovascular mortality by LDL-C-SD

In T2D patients, overall all-cause and cardiovascular mortality rates were lower in patients with LDL-C-SD >10th–≤ 75th percentiles (~20 and ~4/1,000 PY, respectively), and the highest all-cause and cardiovascular mortality rates were consistently found in T2D patients with an LDL-C-SD >90th percentile (~50 and ~10/1,000 PY, respectively; Table S6). T2D patients who took statins had lower all-cause and cardiovascular mortality rates, and those who took fibrates had higher rates of both all-cause and cardiovascular mortality.

Compared to those with an LDL-C-SD in the 25th–50th percentile, patients with an LDL-C-SD <10th or >75th percentile had higher risks of all-cause mortality, and the most elevated HR was observed in those with an LDL-C-SD >90th percentile (Fig. 11). Similarly, among statin users, the highest risks of all-cause mortality were observed in those with an LDL-C-SD <10th or >75th percentile. Among fibrates users, however, higher risks of all-cause mortality were only observed in those with an LDL-C-SD >75th percentile.

T2D patients with an LDL-C-SD in <10th or >90th percentiles had higher risks of cardiovascular mortality (Fig. 12). Among statin users, the highest risks of cardiovascular

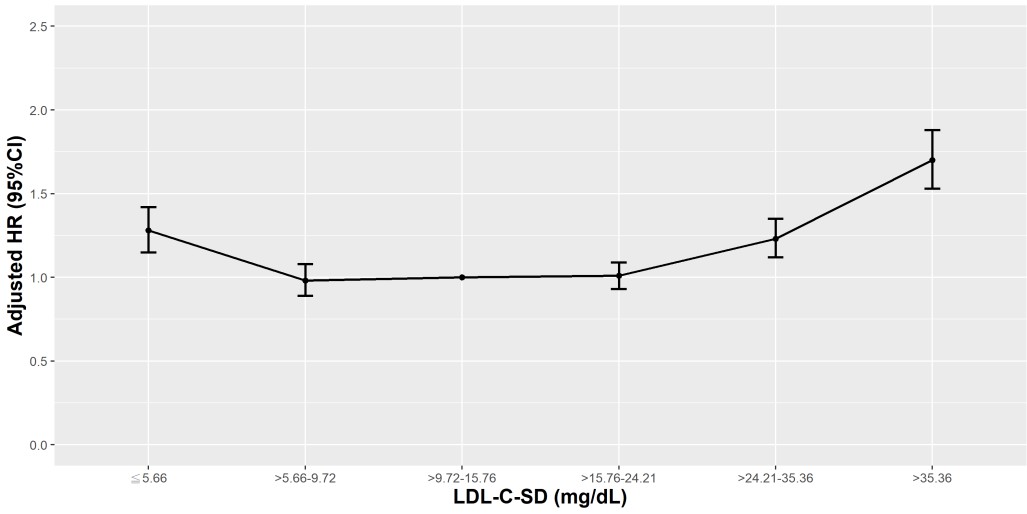

**Figure 11 Hazard plot of low-density lipoprotein cholesterol-standard deviation (LDL-C-SD) and all-cause mortality.**

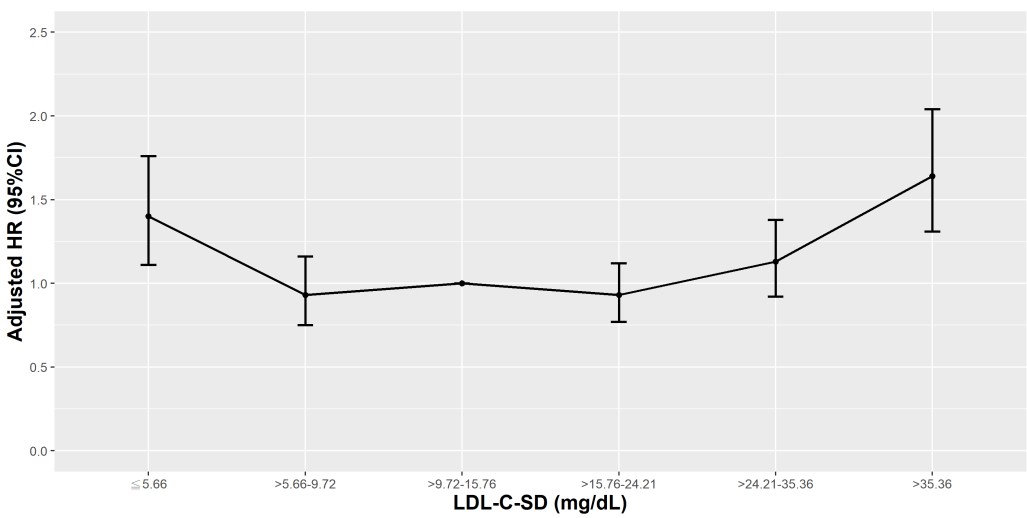

**Figure 12 Hazard plot of low-density lipoprotein cholesterol-standard deviation (LDL-C-SD) and cardiovascular mortality.**

mortality were observed in patients with an LDL-C-SD in the <10th or >90th percentiles. Fibrates use did not impact cardiovascular mortality at any percentile of LDL-C-SD.

## DISCUSSION

In a mean follow-up of about seven years of 46,675 patients with T2D in a tertiary medical center, the lowest all-cause and cardiovascular mortality rates were observed in patients with a mean LDL-C in the 25th–50th percentile, between >90-103.59 mg/dL, with both higher and lower levels of mean LDL-C increasing overall all-cause and cardiovascular

mortality risk in a U-shaped association pattern. A similar pattern was observed in T2D patients of both sexes and among statin users, but the HRs varied by age group. Elevated risks of all-cause mortality persisted in patients with a mean LDL-C ≤77 mg/dL, even after stratification by albumin level. Conversely, an increased risk of cardiovascular mortality was only observed in lower levels of mean LDL-C (<90 mg/dL) in the normal albumin group (≥3.5 g/dL), but no increased risk of cardiovascular mortality was observed in the lower albumin groups (<3.5 g/dL) at any mean LDL-C level.

In general, mean LDL-C levels in people with diabetes are not higher than in people without diabetes matched for age, sex, and body weight (*NCEP, 2002*). Mean LDL-C levels among US patients with diabetes were found to be 106.6 and 110.9 mg/dL for men and women, respectively (*Gu, Kamat & Argulian, 2018*), which are similar to the results of a study of Taiwanese patients with diabetes which found mean LDL-C levels of 106.4 and 106.6 mg/dL for men and women, respectively (*Hsu et al., 2019*).

To the best of our knowledge, the impact of LDL-C levels on all-cause and cardiovascular mortality risk in patients with diabetes has not been studied. However, some results from general population-based studies are in accordance with the findings of our study. A U-shaped association between baseline LDL-C and all-cause mortality risk was observed in the Copenhagen General Population Study, with low and high LDL-C levels associated with an increased risk of all-cause mortality (*Johannesen et al., 2020*); the optimal concentration of LDL-C was found to be 140 mg/dL. Compared to subjects with a baseline LDL-C between 132-154 mg/dL, the multivariate adjusted HR for those with LDL-C <70 mg/dL was 1.25 (95% CI [1.15–1.36]), which was lower than the risk estimates of our study. The authors did not observe any concentration of baseline LDL-C associated with cardiovascular mortality, similar to the results of the low albumin group (<3.5 g/dL) in our study. There was also a lack of an association or an inverse association between all-cause mortality and LDL-C in a systemic review of 19 cohort studies including a total of 68,094 people older than 60 years of age (*Ravnskov et al., 2016*). Similarly, in a Japanese (*Noda et al., 2010*), and a Korean study (*Sung et al., 2019*), lower LDL-C levels were associated with higher all-cause mortality.

There are a few general population-based studies that compare LDL-C levels and mortality risk between sexes. Both NHANES 1994-2014 data from the US (*Liu et al., 2021a*; *Liu et al., 2021b*) and the Ibaraki Prefecture Health Study of Japan (*Noda et al., 2010*) found that the lowest levels of LDL-C (<70 and <80 mg/dL, respectively) increased all-cause mortality risk in both men and women similar to the results of our study. In the Japanese study above (*Noda et al., 2010*), higher, but not lower concentrations of LDL-C were associated with increased mortality risk from coronary heart disease for men, but not for women. In another Korean study, baseline LDL-C levels were not related to cardiovascular mortality in men or women (*Sung et al., 2019*). In our study, however, both men and women with the lowest concentration of mean LDL-C (≤77 mg/dL) had higher risks of cardiovascular mortality. Future research should explore the underlying reasons for these differences in risk between men and women.

Previous population-based studies analyzed the association between all-cause mortality risk and LDL levels in different age groups. In a Korean study of subjects aged 20–39 years

(*Lee et al., 2020*), those with baseline LDL-C levels in the lowest quartile (<84 mg/dL) had the highest risk of all-cause mortality, similar to our results in patients aged <50 years. In a Danish (*Bathum et al., 2013*) and a Japanese study (*Kawamoto et al., 2021*), only those with lower LDL-C had an increased risk of all-cause mortality in subjects aged >50 years. In our analysis, there was a U-shaped relationship between mean LDL-C and all-cause mortality in patients with T2D aged 50-69 years and >69 years. Similar to our findings, another Korean study (*Lee et al., 2022*) found an inverted J-shaped relationship between baseline LDL-C levels and all-cause mortality hazards in individuals aged ≥65 years. There are only a few studies looking at the relationship between mean LDL-C and cardiovascular mortality in various age groups. In a Korean study, there were U-shaped curve associations with LDL-C levels and cardiovascular mortality, but such associations weakened with advancing age (*Yi et al., 2022*). In our research, there were U-shaped and inverse relationships between mean LDL-C and cardiovascular mortality in T2D patients aged 50-69 and >69 years, respectively. In patients aged <50 years, however, mean LDL-C levels were not associated with cardiovascular mortality risk.

Although there is strong evidence that statins reduce total cardiovascular events from many randomized clinical trials, there is less consistency in the magnitude of mortality reduction in patients with diabetes. In our study, in statin users, a mean LDL-C level >90–119 mg/dL was determined as the optimal level with lower and higher levels of mean LDL-C increasing both all-cause and cardiovascular mortality risk. In fibrates users, however, only those with a mean LDL-C ≤77 mg/dL had elevated risks of all-cause mortality, and mean LDL-C levels were not associated with cardiovascular mortality in this cohort.

The relationship between the VVV of LDL-C and all-cause or cardiovascular mortality risk is still contested. Although a Chinese (*Liu et al., 2020*) and a Hong Kong study (*Wan et al., 2020*) revealed that patients with higher quartiles of VVV of LDL-C had marginally increased risks of all-cause mortality, in our analysis, however, lower and higher percentiles of LDL-C-SD were associated with significant all-cause and cardiovascular mortality risk. Similar to our findings, in the Action to Control Cardiovascular Risk in Diabetes (ACCORD) lipid trial, participants with a VVV of LDL-C in the middle deciles had lower all-cause and cardiovascular mortality risk compared to those with a VVV of LDL-C in the first or tenth deciles (*Sheng et al., 2022*). One systemic review and meta-analysis found that mean lipid level might exaggerate the role of lipid variability (*Li et al., 2022*), and not properly adjusting for lipid-lowering medications might be a source of heterogeneity between studies of lipid variability and all-cause mortality. In our analysis, in patients using statins, there was a U-shaped association between LDL-C-SD and all-cause or cardiovascular mortality, but the same association did not exist in patients using fibrates.

Some authors hypothesized that lower cholesterol levels may represent frailty or underlying subclinical disease, and that reduced serum albumin might be used as a surrogate marker of debilitation and severe diseases (*Schupf et al., 2005*). In the Epidemiological Studies in the Elderly (EPESE), *Corti et al. (1997)* observed that the elevated coronary heart disease mortality and all-cause mortality risk seen in those with low baseline TC level (≤160 mg/dL) became insignificant after adjusting for chronic condition like diabetes, hypertension, heart attack, and stroke, together with low serum iron and albumin levels.

In some studies, performed in cardiac intensive units, the nutritional risk index (NRI; (*Lu et al., 2019*) and controlling nutritional status (CONUT) score (*Wang et al., 2021*) were used as baseline indicators of nutritional status. These researchers found that a baseline LDL-C level ≤70 mg/dL was associated with increased incidence of all-cause mortality in the high-risk malnutrition population, but not in the low or moderate groups (*Lu et al., 2019*), and that adjusting for nutritional status eliminated the effects of the lipid paradox (*Wang et al., 2021*).

In contrast, low levels of LDL-C (≤90 mg/dL) were significantly associated with high all-cause mortality risk in both the normal and low albumin group in our study, even after adjusting for coronary risk factors and laboratory results including Hb level. Similarly, increased risks of cardiovascular mortality noted in lower levels of LDL-C (≤90 mg/dL) persisted even after adjusting for covariates in the normal albumin groups, but the statistical significance became null in the low albumin group in the fully adjusted model. Low albumin level is associated with increased mortality (*Touma & Bisharat, 2019*; *Wu et al., 2018*), and in the low albumin group, LDL-C levels did not impact mortality risk. The increased mortality risk seen in low LDL-C levels even after stratifying by albumin level in our study may indicate that low LDL-C may be more than an indirect marker of chronic subclinical disease. An analysis of the Honolulu Program (*Schatz et al., 2001*) revealed that a low serum cholesterol level, maintained over a 20-year period, had the highest risk of all-cause mortality.

The exact mechanisms of the inverse association between LDL-C and mortality are not yet known. From animal and experimental reports, lipoproteins function as part of a nonspecific immune defense system that binds to and effectively inactivates microbes and their toxins through complex formation (*Ravnskov & McCully, 2009*), and high cholesterol may protect against infections (*Weverling-Rijnsburger et al., 2003*) and atherosclerosis (*Ravnskov, 2003*). In one study, TC level was inversely and significantly related to infections in both men and women (*Iribarren et al., 1998*), and a one SD increase in cholesterol was associated with an 8% reduction in the risk of all infections. In another study, an inverse association was found between cholesterol level and pneumonia/influenza hospitalization in both men and women (*Iribarren et al., 1997*). Paradoxically, lower LDL-C levels were associated with in-hospital mortality following acute myocardial infarction (*Al-Mallah et al., 2009*; *Reddy et al., 2015*). Our study results suggest that maintaining very low mean LDL-C level <77 mg/dl long-term may not always confer protective benefits against all-cause and cardiovascular mortality in diabetic patients. Greater VVV of LDL-C might increase the likelihood of plaque vulnerability and rupture leading to instability of the vascular wall and amplification of atherosclerosis (*Sheng et al., 2022*). Previous studies have shown that higher lipid variability may lead to an increased risk of renal function decline, and incidence of end-stage renal disease (*Wan et al., 2020*). Patients with lower or higher LDL-C variability might have displayed worse health conditions that could potentially overshadow the effect of LDL-C variability; further research is warranted to confirm this hypothesis.

However, measurement of LDL-C alone might not be an accurate representation of dyslipidemia. In this study, we could not measure lipoprotein(a) (Lp(a)), an LDL-like

particle consisting of an ApoA moiety linked to one molecule of ApoB 100 (*Qi & Qi, 2012*). The pro-inflammatory, pro-thrombotic, pro-oxidative stress, and pro-atherosclerotic properties of Lp(a) are now recognized as a risk factor for cardiovascular disease (*Wang et al., 2022*) and mortality (*Klingel, Heibges & Fassbender, 2019*). On the basis of observations from multiple laboratories, the cholesterol content of Lp(a) constitutes ∼30–56% of total Lp(a) mass and correcting for Lp(a) in the total LDL-C measurement eliminated LDL-C as a cardiovascular risk factor (*Willeit et al., 2020*). Furthermore, LDL particles are heterogeneous in size and density and can be categorized into large, buoyant LDL (lbLDL) and small, dense LDL (sdLDL). The particles of sdLDL, but not lbLDL, have higher arterial wall penetration, increased susceptibility to oxidation, and are positively associated with major cardiovascular events in patients with diabetes (*Jin et al., 2020*). In hypertriglyceridemia states, excess TG in LDL is hydrolyzed by hepatic lipase and converted to sdLDL. Dyslipidemia in diabetes traditionally refers to an elevation in LDL-C, TG levels, and/or a reduction in HDL-C, but does not usually consider the combination effects of high LDL-C, high Lp(a), and high sdLDL; the limitations of this definition of dyslipidemia in our study might also be a limitation to our analysis.

The strengths of our study are as follows: first, retrieving disease information from the electronic database might have largely reduced both selection and information bias. Second, we assessed the longitudinal records of various laboratory results rather than baseline data which helps prevent regression dilution bias (*Emberson et al., 2003*). Third, we recruited patients using oral or parenteral antidiabetic agents, which might have largely eliminated diabetes misclassification in our study. Fourth, we identified several cardiovascular risk factors, comorbidities, complications, medications, and laboratory results which might have affected the survival of our patients and adjusted for them in our analyses. Most importantly, we considered albumin levels, which might be a surrogate marker for chronic disease, malnutrition, or poor health status in our analysis. Fifth, the Taiwanese population is rather homogenous: more than 95% of Taiwanese people are descendants of immigrants from China, and aborigines or indigenous people comprise only a small percentage of the total population (*Lo et al., 2021*), so there were no race or ethnicity differences among our study subjects.

Several limitations of our study should also be mentioned: we could not detect BMI, smoking status, alcohol consumption, or blood pressure measurements for our study participants, which might have caused residual confounding in our study. However, we did adjust for several cardiovascular risk factors, comorbidities, medications, and laboratory results in our models. We also did not have important health indicators such as frailty, cognitive function, and functional status, but we stratified by albumin level and Hb status of patients with diabetes in our analysis. The ratio of present to ideal body weight was unavailable to assess the NRI together with total lymphocyte count, so we were unable to calculate patient CONUT scores to assess underlying malnutrition status. Furthermore, we could not measure Lp(a) and sdLDL, which are important components of atherogenic dyslipidemia. In addition, this study was based on patients from a single tertiary medical center, and the baseline characteristics of our study participants might have been different

from the general diabetic population, so the generalizability of our study results needs to be confirmed.

## CONCLUSIONS

In conclusion, in a long-term follow-up analysis at a tertiary medical center, low levels of mean LDL-C were associated with increased all-cause and cardiovascular mortality in T2D patients. Increased all-cause mortality risk in patients with lower mean LDL-C levels persisted even after stratifying by albumin level. However, in patients with lower albumin levels, low mean LDL-C level became an insignificant factor for cardiovascular mortality, suggesting baseline health status was a more important factor. The lowest risk of mortality was found in patients with a mean LDL-C >90-103.59 mg/dL. Although statins are recommended for all T2D patients, regardless of LDL-level, more studies in real-world clinical settings are mandatory to re-evaluate the "lower the better lipid hypothesis".

### Funding
This study was supported by grants from the Far Eastern Memorial Hospital (FEMH-2021-C-034, FEMH-2021-C-043 and FEMH-2022-C-017). The funders had no role in study design, data collection and analysis, decision to publish, or preparation of the manuscript.

### Grant Disclosures
The following grant information was disclosed by the authors:
The Far Eastern Memorial Hospital: FEMH-2021-C-034, FEMH-2021-C-043, FEMH-2022-C-017.

### Competing Interests
The authors declare there are no competing interests.

### Author Contributions

- Chin-Huan Chang conceived and designed the experiments, prepared figures and/or tables, authored or reviewed drafts of the article, and approved the final draft.
- Shu-Tin Yeh conceived and designed the experiments, authored or reviewed drafts of the article, and approved the final draft.
- Seng-Wei Ooi conceived and designed the experiments, authored or reviewed drafts of the article, and approved the final draft.
- Chung-Yi Li analyzed the data, authored or reviewed drafts of the article, and approved the final draft.
- Hua-Fen Chen conceived and designed the experiments, analyzed the data, prepared figures and/or tables, authored or reviewed drafts of the article, and approved the final draft.

## Human Ethics

The following information was supplied relating to ethical approvals (i.e., approving body and any reference numbers):

The study was approved by the Institutional Review Board of Far Eastern Memorial Hospital.

## Data Availability

The raw data is owned by Far Eastern Memorial Hospital and is not publicly available because it contains potentially identifying patient information. These restrictions are imposed by the Research Ethics Review Committee of Far Eastern Memorial Hospital according to the Human Subjects Research Act, Laws and Regulations Database, Ministry of Health and Welfare, the Republic of China (Taiwan).

The data is available upon request either through the Department of Medical Research of Far Eastern Memorial Hospital (contact via grace96113@mail.femh.org.tw, +886-2-77284562) or the Research Ethics Review Committee of Far Eastern Memorial Hospital (contact via irb@mail.femh.org.tw, +886-2-77282152, reference number 110276-F) to researchers who meet the criteria for access.

## Supplemental Information

Supplemental information for this article can be found online at http://dx.doi.org/10.7717/peerj.14609#supplemental-information.

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
