# Peer review of "The relationship of low-density lipoprotein cholesterol and all-cause or cardiovascular mortality in patients with type 2 diabetes: a retrospective study"

_PeerJ, doi:10.7717/peerj.14609_

## Round 0.1 · original submission · Major Revisions

As you will see from their comments, all reviewers found this work to be of interest. There are a series of suggestions for changes which I think will make the manuscript more robust and strengthen your conclusions. Reviewer-2 suggests some further statistical analysis, and reviewer-3 suggests that you focus only on people with Type-2 diabetes, and be a little bolder in your assertions.

Please take full cognisance of all comments and address each point in your rebuttal letter, and detail how you have changed your manuscript faced with these comments.

As we do not make a judgement on perceived importance, I am prepared for you to take a pragmatic approach to reviewer-1. His/her point is an important and valid one, and must be carefully and fully discussed in your revised paper, but I will not reject solely on that basis if you carefully address all the other points raised by the other reviewers.

·

Basic reporting

The sole focus on LDL-C in this paper is antiquated. The authors did not address the finding that total LDL-C reported in a conventional lipid panel represents the sum of a heterogeneous population of different LDL particles. One unique population of LDL particles is known as lipoprotein (a) (Lp(a)). Lp(a) is a modified LDL particle in which an apolipoprotein (a) molecule is covalently attached to the ApoB100 moiety of an LDL particle. There is a strong link of Lp(a) levels to CVD, which may be driven by its proinflammatory effects. Thus, high LDL-C, which can be associated with greater mortality, may be based on the inclusion of high Lp(a) levels in the high LDL-C measure. Willeit et al.reported that correcting for the Lp(a) component in the total LDL-C measure eliminated isolated LDL-C as a CVD risk factor.

In addition, investigators have focused on LDL particle types (small dense LDL (sdLDL) and large buoyant LDL (lbLDL)), rather than LDL-C, as a superior measure of CVD risk. The distinction between LDL particle subclasses based on size and density is important because sdLDL is a component of the atherogenic dyslipidemia risk triad, composed of elevated levels of triglycerides (TGs) and sdLDL, in concert with low HDL-C. High TGs, elevated sdLDL and low HDL-C are each, individually, strong markers of CVD risk. Conversely, lbLDL has not been shown to be a CVD risk factor.

Therefore, the focus on LDL-C in relation to total and CVD mortality is of insufficient value to the field.

Reference: Willeit et al. Low-density lipoprotein cholesterol corrected for lipoprotein(a) cholesterol, risk thresholds, and cardiovascular events. J Am Heart Assoc 2020; 9:.

Experimental design

N/A

Validity of the findings

Not meaningful, as discussed in #1 above

Additional comments

It is possible to salvage this manuscript if the authors can provide a more comprehensive review of the literature in the context of LDL particles. It may be that high LDL-C is actually high LDL-C + high Lp(a) + high sdLDL, which may be of value to the field.

Reviewer 2 ·

Basic reporting

The manuscript titled “The relationship of low density lipoprotein cholesterol and al-cause or cardiovascular mortality in patients with diabetes” by Chang et al. describes statistical analysis of LDL-C (Low density lipoprotein cholesterol) levels and relation to patient mortality in over 40,000 patients with diabetes.
The relationship between LDL-C levels and mortality (including cardiovascular mortality) has been extensively studied. As the authors note, these relationships have not been widely reported in the existing literature, in the context of diabetes. The large cohort of diabetic patients carries the novelty of the current study, and the statistical analysis using clinical information collected from them.

Experimental design

1. The authors note that differences in mean LDL-C levels exist between genders, and young vs. old patients (Lines 180-183). Gender-specific stratified statistical analyses, as well as the interaction relationship with age should be explored to answer whether optimal LDL-C levels are identical for the groups, and whether any statistically significant interaction exists.

2. Multiple studies have reported (e.g. PMID: 28071756, PMID: 25122464) that statin treatment in patients with high cholesterol increases their risk for diabetes. Are their differences in glycemic levels and optimal LDL-C in patients undergoing statin treatment vs. fibrates or other drugs?

3. Is the study population homogenous? If not, a breakdown of the demographics (race, ethnicity) should be provided.

Validity of the findings

A recent paper (PMID: 35144636) have reported visit-to-visit variability in LDL-C levels to be a strong predictor of diabetes risk. While mean LDL-C levels may give an approximate measurement of patients . The existence of such statistical relationships can be verifiable in long-term longitudinal patient data, as was collected for this study. If such a relationship does exist, are mean LDL-C levels the most accurate representation of dyslipidemia?
Secondly, is this observed variability significantly associated with the type of antilipid medication?

Additional comments

The authors have performed survival analysis using Cox Hazards Proportional Regression model, and reported the results in terms of Hazard Ratios (Tables 1 and 2). Survival plots and/or hazard ratio event plots should be provided as well to accompany the results in Table1 and Table 2.

·

Basic reporting

I think that Title needs more clarity - at is uses the term diabetes.

This study looked at patients with type I and type II diabetes. These groups are, in many ways, not comparable in various health outcomes, lipid profiles etc. I would recommend the authors take out the type I data (there were only ~ 1% with type I diabetes, and discuss only type II diabetes. [With perhaps a small, separate, table looking only at type I diabetes].

There are a number of language errors.

Just to provide three examples, from the abstract

Line 38. ‘Patients with prescription’ .Missing ‘a’ as in patients with a prescription

Line 38 ‘>6 months in the outpatient visits’ The ‘the is not required.

Line 42 ‘in relation to mean LDL-C in… different albumin levels.’ Requires a statement such as ‘in patient cohorts with different albumin levels

I would recommend finding an English language editor to ensure the English language use is improved. Some of the language errors make understanding difficult.

I think the paper would also be improved with some simple graphics, created from the Tables. The diferent U shaped curves, at different LDL levels in both high and low albumin groups would be highlighted more effectively and would probably make the paper more interesting.

Experimental design

More explanation of use of low albumin

It was not entirely clear why the authors chose a low albumin as their – primary - way of defining poor health/frailty. Nor whether simply having a high, or low, albumin level was sufficient. Whilst LDL-C levels were split into deciles, there were only two albumin groups. High, or low.

I did not find it clear whether or not they had looked at other ways to define poor health/frailty. This could perahps be clarfied.

Validity of the findings

I found myself wondering about the impact of statins on outcomes

It was interesting that statin use was higher in those participants with higher LDL-C levels. 81% in the highest LDL-C group, and 50% in the lowest LDL-C group. I think it would be of value to know the pre-statin LDL-C levels of the participants – if this is known. This would allow the impact of statin treatment to be more accurately evaluated – if possible.

The importance fo this is that all of those with type II diabetes are now advised to take a statin – no matter what their LDL-C level. is If a low LDL-C level carries the greatest burden of overall mortality (especially in those with a low albumin), this advice may need to be altered. Perhaps to ensure that the LDL-C level is maintained within the deciles found to be associated with the lowest overal and/or cardiovascular mortality.

I think more effort to attempt to clarify waht the authors believe tthey found here would be of value.

Additional comments

I think there is also a need to look at the LDL-C levels seen in diabetes (type II mainlly) in Taiwan and relate this to the levels in other countreis e.g. US, UK, etc. I feel this will help to make the paper more relevent to other populations

---

## Round 0.2 · Minor Revisions

We are nearly there. The reviewers agree the manuscript is greatly improved, but a few issues remain about use of English for clarity, and that the discussion overly dense and needs to be clarified and decluttered.

·

Basic reporting

acceptable

Experimental design

acceptable

Validity of the findings

acceptable

Additional comments

the authors have taken my objections into account in the revised manuscript

·

Basic reporting

There are still some issues with the correct use of English. Two examples:

'The optimal levels of low-density lipoprotein cholesterol (LDL-C) in patients with type 2 diabetes (T2D) is not currently clear.' This shoudl be are, as levels referes to two or more.

'we determined the relationship between various LDL-C mean and rates and risks of all-cause or cardiovascular mortality.' I am not sure what this sentence is supposed to mean.

However, it is much improved overall.

Experimental design

I think the experimental design seems fine. The authors acknowledge the fact that certain CV risk factors were not measured, which is perhaps the most important issue here.

Validity of the findings

The findings are valid, for their population, and should be of importance to other, non-Taiwanese populations. The population studies was certainly large enough, for long enough.

Additional comments

I think the disucssion section is very dense. It does make clear that there is much, potentially, conflicting informaiton in this area. It takes some reading.

---

## Round 0.3 · accepted · Accept

Thanks for addressing the remaining issues.